# Biochar Impacts on Acidic Soil from *Camellia Oleifera* Plantation: A Short-Term Soil Incubation Study

**Qianqian Song** [1,2,3]**, Yifan He** [1,2,3]**, Yuefeng Wu** [1,3]**, Shipin Chen** [1,3]**, Taoxiang Zhang** [1,3] **and Hui Chen** [1,3,]*

[1] Forestry College, Fujian Agriculture and Forestry University, Fuzhou 350002, China; song2018@g.ucla.edu (Q.S.); yifanhe20181@g.ucla.edu (Y.H.); yuefengwu01@163.com (Y.W.); fjcsp@126.com (S.C.); xsnzheda2009@163.com (T.Z.)

[2] Department of Molecular, Cell & Developmental Biology, University of California, Los Angeles, CA 90095, USA

[3] Forestry College, Oil Tea Research Center of Fujian Province, Fujian Agriculture and Forestry University, Fuzhou 350002, China

* Correspondence: huichen@fafu.edu.cn; Tel.: +86-139-5034-3791

**Abstract:** Nowadays, biochar is increasingly used widely as an important soil amendment to enhance soil nutrients availability. Therefore, we investigated the effect of *C. oleifera* shell biochar (CSB) on *C. oleifera* plantation soils to provide evidence that *C. oleifera* shell as a raw material in biochar has great potential to be a soil amendment. For this, a short-term incubation experiment was conducted in controlled conditions to evaluate the effects of CSB application on two soil chemical properties, microbial biomass, and enzymatic activity. We compared two acidic soils, mixed with CSB of three pyrolysis temperatures (300, 500, and 700 °C), and two application rates (3% and 5% (*w/w*)), incubated for 180 days. The results showed that the soil pH, total P (TP), and available P (AP) significantly increased under 5CSB700 in two soils, and indicated CSB application rate and pyrolysis temperature had a significant impact on soil pH, TP, and AP ($p < 0.05$). CSB application also significantly increased the total inorganic P in two soils and presented a significantly positive correlation with soil pH, TP, and AP under redundancy analysis. The results suggested that CSB application has a variable effect on soil enzymatic activity, microbial biomass C (MBC), and microbial biomass P (MBP) on average, while it increased the soil microbial biomass N (MBN) in both soils. We concluded that CSB could be a soil amendment to increase soil nutrients of *C. oleifera* plantation soils. Before the application of biochar to *C. oleifera* plantation forest soils, long-term studies are required to assess the effects of biochar under field conditions and its promoting effect on the growth of *C. oleifera*.

**Keywords:** acidic soil; *C. oleifera* shell biochar; microbial properties; soil chemical properties

## 1. Introduction

*Camellia oleifera* Abel., an edible oil tree species belonging to the Theaceae family, originated in the southern area of China and is one of the world's four famous woody oil trees, with over 2000 years of cultivation history [1]. *C. oleifera* oil, which is extracted from the seeds, contains up to 90% unsaturated fatty acid and abundant vitamin E, squalene, flavonoids, as well as tea polyphenols and saponins, which can reduce the risk of cardiovascular disease [2]. In recent years, because of the medicinal value of *C. oleifera* oil and economic values, the demand of *C. oleifera* increased, and "National *C. oleifera* Industry Development Plan (2009–2020)" reported by China's National Development and Reform Commission expected that the area of *C. oleifera* cultivation will increase $1.0 \times 10^6$ ha by 2020. *C. oleifera* is widely cultivated in more than 18 provinces or cities of South China and also distributed in East Asia

and Southeast Asia, in countries such as Japan and Vietnam [3,4]. The ultimate purpose of *C. oleifera* as a tree species is fruit and the yield of fruit is related to soil nutrients. Fujian province is one of the vital *C. oleifera* plantations, where the red and purple soil is the main soil resource with favorable hydrothermal conditions, but also with acidic characteristics and an abundance of hematite iron oxides and low available P content [5]. Moreover, under traditional fertilization management without scientific guidance, such as the excessive use of chemical fertilizers, the soil gradually degraded, and compaction occurred in acidic soil. Thus, *C. oleifera* growth is generally limited.

There are $4 \times 10^6$ ha of *C. oleifera* planting area and yearly fruits yield up to $5.6 \times 10^6$ tons [6]. *C. oleifera* shell (CS), a by-product of *C. oleifera*, accounts for about 50–60% of the whole fruit, as one ton of *C. oleifera* fruit yields 0.54 tons [7,8]. However, there are still more than three million tons of CS normally abandoned or burnt as wastes, leading to a waste of potentially useful resources and pollution of solid waste [7,9,10]. CS contains higher hemicellulose and lignin contents, and thus it can be an alternative raw material for the production of xylose, xylitol, ethanol, and vanillin in the industry [11,12]. So far, most studies of CS mainly focused on the research of biomedical and biologically active components by extracting tea saponin and tannin, the fabrication of aromatic carbon microspheres, as well as analyses of mesopore structural features [13–15].

Biochar is a high carbonaceous solid material produced by the pyrolysis of agricultural and woodsy residual biomass at temperatures ranging from 300 °C to 1000 °C under limited or oxygen-free conditions, and is characterized by a high pH and porous structure, large surface area, and microporous volume, higher thermal stability, and stronger adsorption capacity [16,17]. It has been widely used as soil amendments to acidic and nutrient-poor soil as it can bring major changes in soil structure, water holding capacity, fertility retention, organic phosphorus mineralization, carbon sequestration, and plant growth [18–22]. Moreover, these changes in the soil properties are related to biochar's characteristics and result in changes in enzyme activity and microbial biomass [23–26]. However, biochar application has also been reported to have some negative effects on nutrient availability in acid soils [27]. Therefore, more investigations are needed to provide more evidence for the importance of biochar as a soil amendment. Due to biochar physical and chemical characteristics diversity depending on pyrolysis technology, residence time, temperature, and raw material type, biochar can result in variable effects on soil properties [28,29]. Moreover, the soil type and the application rate of biochar also regulates the effects of biochar on soil [30–33]. Despite the fact that most studies have reported the positive effects of biochar on agricultural systems, such as biochar in maize and rice acting as organic fertilizer by providing soil nutrients that were present in the precursor biomass, fewer details are reported about the forest soils [34–36]. Furthermore, some studies report biochar as a new method for the reuse of CS, while other reports have used CS as raw material to produce biochar, but only study the component, properties, and structure [37,38]. Therefore, a further study of the effects of biochar pyrolyzed from CS on *C. oleifera* plantation forest soil is needed before CSB is used as a soil amendment.

This study aimed to examine how CSB affects the soil chemical properties, P fractionation, microbial biomass, and enzyme activities in *C. oleifera* plantation forest soil, in order to assess comprehensively and understand better the effect of CSB on the soil. Thus, this study can inform future *C. oleifera* plantation soil management practices and lay a foundation for the recycle use of *C. oleifera* waste residue.

## 2. Materials and Methods

### 2.1. Soil and Biochar Preparation

The soil was collected to a depth of 40 cm in Fujian province, China: red soil (RS) and purple soil (PS). Two soil samples with acidic pH were collected from over 10-year-old *C. oleifera* plantations, located in Fuzhou (RS, 26°15′59″ N, 119°23′62″ E) and Ninghua (PS, 26°19′35″ N, 116°47′21″ E). The red soil and purple soil samples were air-dried and then crushed to pass through a 2.0 mm sieve before conducting the incubation experiment, respectively. The initial properties of soils were determined and are shown in Table 1.

The biochar was produced from CS collected in Fuzhou, Fujian Province, China, the main *C. oleifera* plantation. Air-dried CS was pyrolyzed in a laboratory furnace (SSDP-5000-A, Nanjing, Jiangsu, China) at 300, 500, and 700 °C, respectively, under an oxygen-free condition with 2 h residence time to produce different kinds of biochar (henceforth CSB300, CSB500, and CSB700, respectively). The CSB samples were crushed to pass through a 0.149-mm sieve before it was applied to the soils. The initial properties of CSB samples were measured and are shown in Table 1.

The pH value was measured in a 1:2.5 soil-water suspension by glass membrane electrode (PE-10, Sartorius, Goettingen, Germany). Total C and N were determined by CNS Analyzer (Vario EL III, Elementar, Frankfurt, Hesse, Germany). The soil samples were digested with $H_2SO_4$-$HClO_4$ to measure total P and other elements using ICP atomic emission spectroscopy (ICP-MS-2030, Shimadzu, Kyoto, Japan). Available P was extracted by 0.5 M $NaHCO_3$ and measured by the molybdenum blue colorimetric method [39]. The organic matter was measured by dry combustion using a CNS Analyzer (Vario EL III, Elementar, Frankfurt, Hesse, Germany).

**Table 1.** Initial properties of two soil types and three different pyrolysis temperature CSB (g·kg$^{-1}$).

|  | **RS** | **PS** | **CSB300** | **CSB500** | **CSB700** |
|---|---|---|---|---|---|
| Ash | - | - | 9.37 | 15.01 | 21.78 |
| Yield | - | - | 43.16 | 32.67 | 19.24 |
| pH | 4.29 | 4.74 | 6.17 | 10.14 | 11.82 |
| TC | 7.45 | 8.65 | 378.9 | 545.7 | 671.6 |
| TN | 0.44 | 0.55 | 5.44 | 3.71 | 2.49 |
| TP | 0.21 | 0.24 | 1.69 | 3.09 | 6.02 |
| K | 12.47 | 20.19 | 9.44 | 11.46 | 18.53 |
| OM | 9.23 | 13.85 | - | - | - |
| AP | 0.006 | 0.008 | - | - | - |
| Fe | 216.8 | 114.7 | 2.37 | 2.54 | 2.51 |
| Al | 7.29 | 4.51 | 1.55 | 1.63 | 1.47 |
| Ca | 3.96 | 4.08 | 2.29 | 4.67 | 3.51 |
| Na | 5.12 | 5.29 | 0.29 | 0.37 | 0.64 |
| Mg | 2.04 | 2.17 | 2.44 | 3.18 | 4.86 |
| Mn | 0.54 | 0.43 | 0.63 | 0.72 | 0.71 |
| S | 0.29 | 0.32 | 0.54 | 0.38 | 0.22 |
| Texture (%) | | | | | |
| Clay | 34.10 | 27.69 | - | - | - |
| Sand | 47.10 | 50.94 | - | - | - |
| Silt | 18.80 | 21.37 | - | - | - |

Ash (%) = ($m_2$/$m_1$) × 100%, $m_1$ is biochar weight and $m_2$ is residue constant weight after heated at 750 °C for 2 h. Yield (%) = (biochar weight/*C. oleifera* shell weight) × 100%. Abbreviations: TC = total C, TN = total N, TP = total P, OM = organic matter, AP = available P ($n$ = 3). "-" represents not measured.

### 2.2. Soil Incubation Experiment

Initially, two type air-dried soils (RS and PS) were rewetted to 60% moisture content, and pre-incubated at constant 25 °C for 7 days, respectively. Then, the 2.0 kg of pre-incubated soil was weighed into ziplock bags (240 × 340 mm), mixed with CSB produced at three pyrolysis temperatures (300 °C, 500 °C, and 700 °C) under 3% (*w/w* 60 g bag$^{-1}$) and 5% (*w/w* 100 g bag$^{-1}$) application rates (henceforth 3CSB300, 3CSB500, 3CSB700, 5CSB300, 5CSB500, and 5CSB700, respectively), and thoroughly incubated for 180 days.

The factorial experimental design for analyzing soil chemical properties, enzymatic activity, and microbial biomass involving twelve CSB treatments and the unamended soil control (RS and PS) treatments are described as follows: red soil with 3% application rate CSB (henceforth RS + 3CSB300, RS + 3CSB500, RS + 3CSB700), red soil with 5% application rate CSB (henceforth RS + 5CSB300, RS + 5CSB500, RS + 5CSB700), purple soil with 3% application rate CSB (henceforth PS + 3CSB300, PS + 3CSB500, PS + 3CSB700) and purple soil with 5% application rate CSB (henceforth PS + 5CSB300,

PS + 5CSB500, PS + 5CSB700). The incubation experiment was carried out in a constant temperature incubator with 25 °C with a water concentration of 60% of field water holding capacity of the soil by adjusting the water mass every 3 days by weighing the loss throughout the incubation experiment.

### 2.3. Soil Characteristics Analysis

After the 180-day incubation experiment, all subsamples of soil were divided into two parts. One part was air-dried, crushed, and sieved before to the determination of soil chemical properties, and the other part was stored at 4 °C for the analysis of microbial properties.

These soil samples were used to determine soil pH value, OM, TC, TN, TP, and AP. The P fractionations of soil were measured by the molybdate-blue method at a wavelength of 700 nm [40,41]. Briefly, 1.00 g soil was weighed into 50 mL centrifuge tubes and $NaHCO_3$ P was extracted by 30 mL of 0.5 M $NaHCO_3$ (pH 8.5), NaOH P by 20 mL of 0.1 M NaOH, and HCl Pi by 30 mL of 1 M HCl. All extraction samples were shaken for 16 h at 20 °C, centrifuged at 4000× $g$, and the supernatant collected. Residual P in soil samples was extracted by digestion with $H_2SO_4$. All extracts were determined with the molybdate-blue method at a wavelength of 700 nm. Soil samples were done three replications.

Soil microbial biomass C (MBC), microbial biomass N (MBN), and microbial biomass P (MBP) were measured by the chloroform fumigation extraction method [42–44]. Briefly, 25.00 g of each soil sample was fumigated for 24 h in a glass desiccator at 25 °C with 30 mL ethanol-free $CHCl_3$. Weight another 25.00 g of each soil sample for non-fumigation. For MBC, weight 10.00 g fumigated and non-fumigated soil samples, respectively, and extracted with 50 mL 0.5 M $K_2SO_4$ for 30 min and centrifuged. For MBN, weight another 10.00 g soil sample from fumigated and non-fumigated soil samples, respectively, and extracted with 40 mL 0.5 M $K_2SO_4$ for 30 min and centrifuged. The supernatant was measured for dissolved organic carbon (DOC) and TN by a TOC analyzer (TOC-L, Kyoto, Japan). The soil MBC = (DOC fumigated – DOC non-fumigated) × 0.45. The soil MBN = (TN fumigated – TN non-fumigated) × 0.54. For MBP, weight 2.00 g fumigated and non-fumigated soil samples, respectively, and extracted with 50 mL 0.5 M $NaHCO_3$ for 30 min and centrifuged. The supernatant was measured for soil MBP by the molybdate-blue method at a wavelength of 700 nm. The soil MBP = (P fumigated – non-fumigated) × 0.40. Soil enzyme activities of invertase and catalase were assayed following the methods by Guan [45]. Acid phosphatase activity was assayed from the amount of p-nitrophenol released by measuring the absorbance at 410 nm [46]. Urease activity was estimated according to the determination of $NH^{4+}$-N remaining by taking the absorbance at 690 nm [47].

### 2.4. Statistical Analysis

All statistical analyses were conducted by IBM SPSS Statistics 20.0, and the significant differences of all treatments were evaluated by a one-way ANOVA analysis ($p < 0.05$). Effects and interactions of soil type, biochar pyrolysis temperature, and biochar application rate on soil chemical and microbial properties were analyzed by three-way ANOVA. The correlation coefficients between the chemical properties and soil enzyme activities were analyzed by Pearson's nonparametric test. Subsequently, a redundancy analysis (RDA) was conducted to analyze the relationship between the soil P fractionation and chemical properties in R. A principal component analysis (PCA) was conducted to analyze the changes of soil under different CSB treatments on soil microbial biomass and enzyme activity.

## 3. Results and Discussion

### 3.1. Characteristics of CSB

The physical-chemical properties of CSB were presented in Table 1. Using *C. oleifera* shell as raw material to produce biochar by pyrolysis, the ash of the resulting biochar showed that, with the increase in pyrolysis temperature, the ash content is proportional to the temperature change and the 9.37–21.78% ash content in this incubation study. Higher ash content represented the more loss of organic matters during the process of pyrolysis and the residuals are the higher crystalline mineral

elements which can be a nutrient source of biochar treated soil [48–50]. Qin et al. [37] reported that the yield of CSB was reduced with the increase in carbonization temperature (from 53.25% to 30.62%), and this result was consistent with those of the present study (from 43.16% to 19.24%). The decrease of biochar yield was mainly attributed to the organic materials decomposing during the constant pyrolysis process. In our study, biochar was weakly acidic or alkaline (pH value between 6.17 and 11.82), and the pH values increased with the increase in pyrolysis temperature. Most studies had indicated that the pH value of biochar increased by acidic substance and alkaline metals gradually volatilized and accumulated in material with the increase in pyrolysis temperature [51,52].

Comparing the elemental content of biochar pyrolysis at three different pyrolysis temperatures, it was showed that the TC was the highest, and increased with the increase in pyrolysis temperatures, similar results also found in Qin et al. [37]. Moreover, Fan et al. [38] compared the characterization of CSB with raw material and showed that the TC is increased after pyrolysis at 600 °C from 47.21% to 73.03%. In the present study, TP and K also increased with an increase in pyrolysis temperature. Dieguez-Alonso et al. [53] showed that the accumulation of TP and K content may due to the release of volatile organic matter during the pyrolysis process and shows a dependency on pyrolysis temperature. Hossain et al. [54] concluded that pyrolysis could lead to a loss of nitrogen compounds in the raw material due to nitrogen content volatilization during the pyrolysis process. This may explain the fact that the TN of CSB decreased obviously with the increase in pyrolysis temperature. The content of Fe and Al in different CSB with no significant variations during the pyrolysis process, changes from 2.37 to 2.54 g kg$^{-1}$ and from 1.47 to 1.63 g kg$^{-1}$, respectively, and only a slight part of these micro-nutrients were lost with increasing temperature. Moreover, the other micro-nutrient of Na, Mn, and S content in CSB was relatively low and showed variable changes with the pyrolysis temperature. Overall, these high nutrient concentrations in biochar confirmed that biochar could have great potential as a soil amendment in the soil, as in previous studies [55].

### 3.2. CSB Effects on Soil Chemical Properties

After 180 days of soil incubation with CSB, the pH value in all CSB-amended soils increased significantly ($p < 0.05$) with the increase in CSB application rate and pyrolysis temperature (Table 2). All CSB treatments increased soil pH value in the red soil by 0.59-1.84 units compared with RS and followed the order: RS + 5CSB700 > RS + 5CSB500 > RS + 3CSB700 > RS + 3CSB500 > RS + 5CSB300 > RS + 3CSB300 > RS, and increased soil pH in the purple soil by 0.78–1.46 units compared with PS, followed the order: PS + 5CSB700 > PS + 5CSB500 > PS + 5CSB300 > PS + 3CSB700 > PS + 3CSB500 > PS + 3CSB300 > PS. Under the treatment of 5CSB700, the highest pH value of soil was achieved in both soils, while the lowest pH value was observed in the treatment of 3CSB300. Moreover, the three-way ANOVA analysis showed that soil type, biochar pyrolysis temperature, and application rate had significant effects on soil pH value, and the interaction of soil type and biochar pyrolysis temperature also had a significant effect on soil pH value.

Several studies have shown soil pH value generally increases after biochar applied in different soil types, especially in acid soils [56–58]. However, some studies found that biochar application by weight as a soil amendment showed no significant increase in soil pH value [59,60]. In this paper, all CSB amendments caused significant increases in both soil pH value, and the pH value increased significantly with the increase in CSB application rate and pyrolysis temperature (Table 2). After CSB was applied to the soil, alkaline material of CSB was released, leading to an increased soil pH value, especially under CSB700 treatment. Meanwhile, biochar contained rich base ions of Mg, Na, and Ca, which reduced active cations and consumed protons to improve pH value in the soil [27,61]. Peng et al. [30] showed that, with the increase of pyrolysis temperature, biochar obtained rich alkaline functional groups and ash content. Therefore, the soil amended with biochar produced at high pyrolysis temperature was better than that produced at lower pyrolysis temperature.

**Table 2.** Red and purple soil/biochar mixtures properties after incubation of the experiment. Three-way ANOVA for chemical properties measured in both soils. The table shows *P* value for the three factors: soil type (ST), biochar pyrolysis temperature (BPT) and biochar application rate (BAR).

| Treatment | pH (1:2.5) | OM($g \cdot kg^{-1}$) | TC($g \cdot kg^{-1}$) | TN ($g \cdot kg^{-1}$) | TP ($g \cdot kg^{-1}$) | AP ($g \cdot kg^{-1}$) |
|---|---|---|---|---|---|---|
| RS | 4.18 ± 0.06e | 10.89 ± 0.47e | 7.27 ± 0.09c | 0.37 ± 0.04c | 0.236 ± 0.01e | 0.0056 ± 0.00d |
| RS + 3CSB300 | 4.77 ± 0.04d | 14.92 ± 0.44b | 21.97 ± 3.10b | 0.46 ± 0.17bc | 0.347 ± 0.02d | 0.0103 ± 0.00c |
| RS + 3CSB500 | 5.27 ± 0.08c | 13.08 ± 0.65d | 21.46 ± 4.20b | 0.49 ± 0.09abc | 0.387 ± 0.03cd | 0.0118 ± 0.00bc |
| RS + 3CSB700 | 5.41 ± 0.02b | 13.35 ± 0.60cd | 24.00 ± 1.73b | 0.53 ± 0.13abc | 0.483 ± 0.05ab | 0.0188 ± 0.00a |
| RS + 5CSB300 | 5.14 ± 0.04c | 16.83 ± 0.93a | 27.18 ± 2.33b | 0.58 ± 0.01abc | 0.386 ± 0.02cd | 0.0144 ± 0.00b |
| RS + 5CSB500 | 5.53 ± 0.03b | 13.51 ± 0.08cd | 35.84 ± 0.99a | 0.67 ± 0.07ab | 0.423 ± 0.05bc | 0.0144 ± 0.00b |
| RS + 5CSB700 | 6.02 ± 0.12a | 14.40 ± 0.17bc | 40.88 ± 2.73a | 0.70 ± 0.09a | 0.529 ± 0.02a | 0.0195 ± 0.00a |
| PS | 4.62 ± 0.05e | 12.23 ± 0.33c | 9.39 ± 0.51b | 0.69 ± 0.01bc | 0.242 ± 0.01e | 0.0077 ± 0.00c |
| PS + 3CSB300 | 5.40 ± 0.00d | 16.87 ± 1.03a | 19.35 ± 4.37a | 0.81 ± 0.04a | 0.299 ± 0.00b | 0.0096 ± 0.00b |
| PS + 3CSB500 | 5.59 ± 0.03c | 14.05 ± 1.01b | 22.40 ± 3.24a | 0.67 ± 0.04cd | 0.321 ± 0.04b | 0.0100 ± 0.00b |
| PS + 3CSB700 | 5.60 ± 0.13c | 14.43 ± 0.94b | 22.54 ± 2.53a | 0.70 ± 0.03bc | 0.326 ± 0.00b | 0.0111 ± 0.00b |
| PS + 5CSB300 | 5.70 ± 0.07bc | 17.42 ± 0.67a | 25.43 ± 3.89a | 0.68 ± 0.01c | 0.336 ± 0.02b | 0.0132 ± 0.00a |
| PS + 5CSB500 | 5.79 ± 0.06b | 14.28 ± 1.07b | 27.38 ± 4.95a | 0.75 ± 0.03b | 0.388 ± 0.01a | 0.0132 ± 0.00a |
| PS + 5CSB700 | 6.08 ± 0.13a | 14.45 ± 0.30b | 27.51 ± 4.48a | 0.61 ± 0.02d | 0.408 ± 0.00a | 0.0108 ± 0.00b |
| ST | <0.001 | 0.006 | 0.005 | <0.001 | <0.001 | <0.001 |
| BPT | <0.001 | <0.001 | 0.022 | 0.823 | <0.001 | <0.001 |
| BAR | <0.001 | 0.029 | <0.001 | 0.090 | <0.001 | <0.001 |
| BPT * BAR | 0.001 | 0.457 | 0.315 | 0.080 | 0.064 | 0.090 |

Data means ± SD (*n* = 3); Different letters within a line indicate significant differences between treatments.

It is crucial to maintain a suitable content of soil organic matter and nutrients cycling for any soil management. The decrease in soil organic matter generally is accompanied by a reduction of soil fertility, and leading to some soil degradation problems as erosion, biodiversity loss nutrient deficiency, and salinization. The results showed that CSB applied in the soil caused significant increases in the organic matter in both soils after 180-day lab incubation (Table 2). Moreover, 5CSB300 were enhanced significantly (*p* < 0.05) compared with RS and PS, and higher than that in the treatment of control by 5.94 and 5.19 units, respectively, followed by 3CSB300. However, no matter what application rate of biochar applied, there was no significant difference in CSB500 and CSB700 in both soils. Moreover, although soil type, biochar pyrolysis temperature, and application rate had significant effects on soil OM, respectively, no effect of the interaction of three factors were not reported.

The highest total C level in the red soil was observed in the 5CSB700, and the same result was also found in the purple soil. Further, TC in all CSB treatments was significantly higher (*p* < 0.05) than that in the RS and PS by 14.7–33.61 and 9.96–18.12 units, respectively. However, there was no difference was shown between RS + 5CSB300 and the CSB treatments under 3% application rate in the red soil. Further, there was also no significant difference observed among all CSB-amended purple soil treatments. All CSB treatments had higher TC amounts as compared to the RS and PS, indicating that CSB application could increase soil TC, since all CSB amendments contained more than 370 $g \cdot kg^{-1}$ TC, which can provide a large amount of C to soil and increase C stock. Similar results verified that the application of biochar to the soil increased the TC with the increase application rate [62,63].

Compared with RS, RS + 5CSB700 observed the highest value of total N content among all biochar applications, increased significantly (*p* < 0.05) by 1.89 times, followed by RS + 5CSB500 (1.81 times), but other CSB treatments have no significant difference with RS. Among all the treatments, PS + 3CSB300 obtained more TN content (0.81 $g \cdot kg^{-1}$) as compared to other treatments, and PS, PS + 3CSB700, and PS + 5CSB500 also increased, but the difference was not significant with PS. Although PS + 3CSB500 and PS + 5CSB300 decreased the soil TN, no difference with PS. However, PS + 5CSB700 decreased TN of purple soil significantly, decreased by 13.11%, compared to PS. Moreover, as shown for single factors, pyrolysis temperature, rate of CSB application, and soil type showed a significant effect on soil TC and TN, respectively. This short-term incubation study showed that the TN contents in the red soil were increased with CSB application rate and pyrolysis temperature, respectively, which is consistent with previous results [62,64]. Sackett et al. [65] also found that the biochar application had no effect

on soil. However, the increase of TN in both soils with biochar application may be caused by the N content of biochar released into the soil and retains as a stable form [63,66].

The soil total P levels after CSB applied for 180-day in the red and purple soil increased with increasing application rate and pyrolysis temperature of CSB, a similar trend also found in available P of the red soil (Table 2). TP in the CSB treatments was higher than RS and PS, increased significantly by 110.68–293.44, and 56.83–166.20 units, respectively. Also, soil type, biochar pyrolysis temperature, and application rate had significant effects on soil TP, and the interaction of soil type and biochar pyrolysis temperature also had a significant effect on soil TP. The highest AP value in the red and purple soil was RS + 5CSB700 and PS + 5CSB500, which increased by 247.07% and 71.21%, respectively, compared to PS. In this incubation study, the red soil and purple soil both was tropical soil of phosphorus deficiency, the results showed that TP and AP also significantly increased after the application of the CSB, particularly at the application rate of 5% with the same pyrolysis temperature, and higher pyrolysis temperature CSB obviously increased the soil TP than lower pyrolysis temperature. Our study also showed that TP and AP increased significantly with the increases of CSB application rate in both soils. The application of biochar could increase soil TP content by the biochar itself high P retention ability and also could increase the P adsorption capacity by its high specific surface area and abundant functional groups [67,68]. The previous study also reported that there was a high TP and AP content on biochar surface contributed to soil directly leading to the increasing TP and AP [69].

### 3.3. CSB Effects on Soils P Fractionations

Changes in soil sequential P fractionation of two type soils are shown in Table 3. In the red soil, compared with RS, resin P increased in all CSB treatments and the highest increase was observed in RS + 5CSB500, by 6.11 units, compared to RS. Although RS + 3CSB300 increased resin P level by 0.28 units than RS, while no significant differences with RS. As for the purple soil, PS + 3CSB300 decreased the resin P level by 0.03 units, while other CSB treatments significantly increased resin P level by 3.19–7.93 units, compared to PS. 5CSB700 which induced the most significant ($p < 0.05$) effect on $NaHCO_3$ P contents, increasing from 12.79 $mg·kg^{-1}$ and 16.35 $mg·kg^{-1}$ under RS and PS, respectively, to 23.55 $mg·kg^{-1}$ and 43.67 $mg·kg^{-1}$ after CSB was applied. There was no clear effect of RS + 3CSB300 on the red soil $NaHCO_3$ P, but the other CSB treatment had a large variation on $NaHCO_3$ P, compared to RS and PS.

**Table 3.** Phosphorus fractionations of two soil types after incubation of different pyrolysis temperature CSB application ($mg·kg^{-1}$).

| Treatment | Resin P | $NaHCO_3$ $P_o$ | $NaHCO_3$ $P_i$ | NaOH $P_o$ | NaOH $P_i$ | HCl $P_i$ |
|---|---|---|---|---|---|---|
| RS | 3.90 ± 0.21d | 5.35 ± 0.32d | 7.44 ± 1.80e | 69.02 ± 2.16a | 25.23 ± 1.87d | 16.40 ± 1.28e |
| RS + 3CSB300 | 4.18 ± 0.15d | 6.25 ± 0.10cd | 7.81 ± 0.24e | 66.84 ± 2.90ab | 27.12 ± 0.89d | 18.78 ± 0.44d |
| RS + 3CSB500 | 6.74 ± 0.18c | 6.97 ± 0.65c | 13.11 ± 0.40b | 45.37 ± 2.91d | 40.66 ± 0.65a | 24.21 ± 0.16a |
| RS + 3CSB700 | 7.95 ± 0.33b | 8.74 ± 0.31ab | 12.83 ± 0.23b | 56.92 ± 1.80c | 37.55 ± 1.11b | 21.80 ± 0.32bc |
| RS + 5CSB300 | 6.85 ± 0.76c | 7.13 ± 0.09c | 10.64 ± 0.34d | 58.50 ± 2.02c | 32.25 ± 0.26c | 22.97 ± 0.23b |
| RS + 5CSB500 | 10.01 ± 0.52a | 7.84 ± 0.71bc | 11.27 ± 0.30c | 60.22 ± 2.65bc | 33.32 ± 1.83c | 20.77 ± 0.24c |
| RS + 5CSB700 | 9.99 ± 0.48a | 9.07 ± 0.57a | 14.48 ± 0.17a | 47.02 ± 7.79d | 43.20 ± 0.98a | 24.54 ± 0.14a |
| PS | 5.88 ± 0.34d | 8.13 ± 0.35e | 8.22 ± 1.15f | 60.79 ± 2.72a | 21.87 ± 0.65f | 19.95 ± 1.23b |
| PS + 3CSB300 | 5.85 ± 0.30d | 15.78 ± 1.04d | 13.03 ± 0.22e | 53.93 ± 1.86abc | 30.22 ± 3.99e | 23.00 ± 0.96ab |
| PS + 3CSB500 | 12.90 ± 0.86b | 19.36 ± 1.58c | 14.40 ± 0.25d | 54.01 ± 3.85ab | 37.50 ± 2.03cd | 21.15 ± 2.86b |
| PS + 3CSB700 | 13.15 ± 0.23ab | 21.37 ± 0.96c | 17.38 ± 0.11b | 49.48 ± 2.94bc | 41.20 ± 3.99bc | 19.21 ± 2.45b |
| PS + 5CSB300 | 9.07 ± 1.24c | 20.11 ± 1.13c | 13.44 ± 0.08de | 56.05 ± 6.15ab | 31.92 ± 1.33de | 23.08 ± 1.09ab |
| PS + 5CSB500 | 13.81 ± 0.60a | 23.22 ± 1.09ab | 15.88 ± 0.10c | 48.50 ± 1.48c | 45.91 ± 2.09ab | 25.20 ± 1.76a |
| PS + 5CSB700 | 13.51 ± 0.59a | 24.24 ± 0.76a | 19.43 ± 0.13a | 46.67 ± 2.63c | 47.23 ± 3.25a | 25.10 ± 1.72ab |

Abbreviations: $P_o$ = organic phosphorus, $P_i$ = inorganic phosphorus. Data are means ± SD ($n = 3$), different letters within a line indicated significant differences ($p < 0.05$) between treatments.

The NaOH Po in the red soil decreased under the CSB amendment, dropped from 69.02 $mg·kg^{-1}$ to 45.37 $mg·kg^{-1}$. Moreover, the NaOH Po in purple soil also decreased, dropped by 4.74–14.12units.

Meanwhile, the NaOH Pi of the red soil under the CSB application generally increased by 1.89–17.97 units, compared to RS. The peak NaOH Pi levels were observed with RS + 5CSB700 and PS + 5CSB700 in red and purple soil, respectively, while no significant difference was observed with each other. The HCl Pi level of the red soil with CSB applied was significantly higher in RS + 3CSB500 and RS + 5CSB700 than RS, while other CSB treatments all had varied increases in HCl Pi values. In contrast to PS, a significant increase in the HCl Pi values by 5.25 units was observed in PS + 5CSB500, and there were little variations between PS and other CSB treatments. Generally, there was a significant difference ($p < 0.05$) between the organic and inorganic P pool after 180 days incubation with CSB (Table 3).

An increase in available P pools (resin Pi and NaHCO$_3$ Pi) was observed in our study after CSB applied into both soils, and the resin Pi and NaHCO$_3$ Pi were available forms of soil P that could be easily taken up by plants and microorganisms [70]. Moreover, in the present study, the 3CSB300 treatment had no significant effect on the increase of resin Pi and NaHCO$_3$ Pi, compared to RS and PS. Some studies reported that the acid soils with higher P sorption capacity were due to the presence of high amounts of free Al and Fe oxides, and NaOH P generally bound to Fe and Al oxides, thus becoming a non-available form of soil P (NaOH Po) [19,70,71]. Further, 5CSB700 treatment in the red soil demonstrated a significant decrease in the content of NaOH Po more than other CSB treatments and RS, and a decreasing tendency was also found in the purple soil. In the present study, an increase in soil P availability of NaOH Pi while decreased in NaOH Po after 180-day CSB treated soil was found in the present study in both soils, which may cause by P sorption to Al and Fe decreased. As a mainly insoluble and stable form of P, HCl P was regarded as a non-labile form of P pool in the soils, while HCl Pi was supposed to be sparingly available pools of P, the increase in the content of HCl Pi in this study demonstrated that CSB treatment had a positive effect on soil inorganic P [72].

### 3.4. Relationship between Soil Chemical Properties and P Fractionations after 180-Day Incubation

The solubility of the various organic and inorganic P directly influenced by the soil pH value and becomes more available in soil within the pH value range 6.5–7.5 [27]. In our study, the relationship among the soil selected chemical properties (pH value, TP, and AP) and different soil P fractionations analyzed by the redundancy analyses (RDA) as shown in Figure 1. The first (RDA1) and second (RDA2) ordination axes accounted for 93.36% and 6.19% of the total variation in the red soil between RS and CSB treatments of 3% application rate, respectively (Figure 1A), AP and TP both were strongly positively correlated with the content of resin P and HCl Pi. Moreover, the pH value was also significantly positively correlated with the content of NaHCO$_3$ Po, NaOH Po, and NaHCO$_3$ Pi. While the content of NaOH Po was significantly negatively correlated with the above properties. Moreover, there was a similar tendency in the red soil under 5% application rate treatments as under 3% application rate treatments, the first two RDA axes explained 95.77% and 3.58% of the total variation in Figure 1B, respectively. However, the NaOH Po showed an opposite trend, decreasing with the application of CSB and showing a positive correlation with soil TP, AP, and pH value in the red soil. In the purple soil, under 3% application rate treatments, the first two RDA axes explained 96.99% and 2.58% of the total variation in the data, respectively (Figure 1C), and under 5% application rate treatments, the first two RDA axes explained 95.24% and 4.06% of the total variation in the data, respectively (Figure 1D). Moreover, AP showed a strongly negatively correlated with the content of NaOH Pi and NaOH Po in the purple soil, and AP was also significantly negatively correlated with the content of HCl Pi.

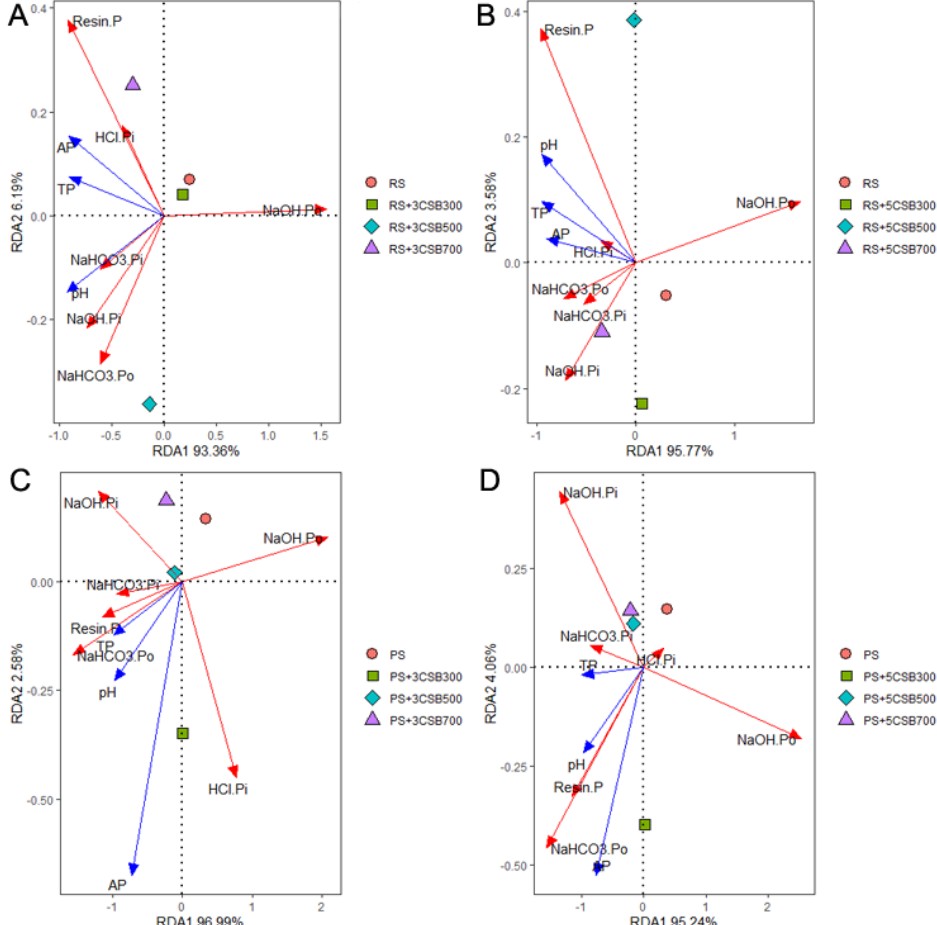

**Figure 1.** Redundancy analyses (RDA) of the correlations between soil chemical properties and P fractionation in all treatments. (**A**): RDA of red soil with 3% application rate of *C. oleifera* shell biochar (CSB); (**B**): RDA of red soil with 5% application rate of CSB; (**C**): RDA of purple soil with 3% application rate of CSB; (**D**): RDA of purple soil with 5% application rate of CSB; The blue arrows indicate the soil chemical properties had a significant effect on soil inorganic P fractions ($p < 0.05$).

*3.5. CSB Effects on Soils Microbial Properties*

Other than chemical properties, the application of biochar applied in the soil also induced changes in microbial activity. Previous studies reported that the effect of biochar on soil enzyme activities mainly depended on biochar application rate, pyrolysis temperature [66,73]. Some studies reported that biochar application produced at high temperature ($\geq$400 °C) decreased soil enzyme activity, which may be explained by the fact that biochar sorbs the enzyme activity and a substrate non-selectively [74]. At the end of the incubation experiment, the activities of invertase, urease, acid phosphatase, and catalase were modified by the CSB application (Figure 2).

In the red soil, 3CSB300 and 5CSB500 both significantly increased invertase activity, increased by 0.38 and 0.11 units, respectively, compared to control (Figure 2A). Although 3CSB500 and 5CSB500 both increased invertase activity, no significant difference with control in the red soil. However, 3CSB700 showed a significant negative effect on soil invertase activity, which decreased by 0.16 units. As for purple soil, all biochar treatments had significant negative effects on soil invertase activity, decreased by 0.23–0.61 units, compared to control. In our study, biochar application rate, pyrolysis temperature showed different effects on enzyme activity (Figure 2 and Table 4). Our study also shows that CSB700 as a soil amendment inhibited soil invertase activity, other CSB treatments increased the activity of invertase more or less in the red soil, while all biochar applied treatments significantly suppressed invertase activity in purple soil (Figure 2A).

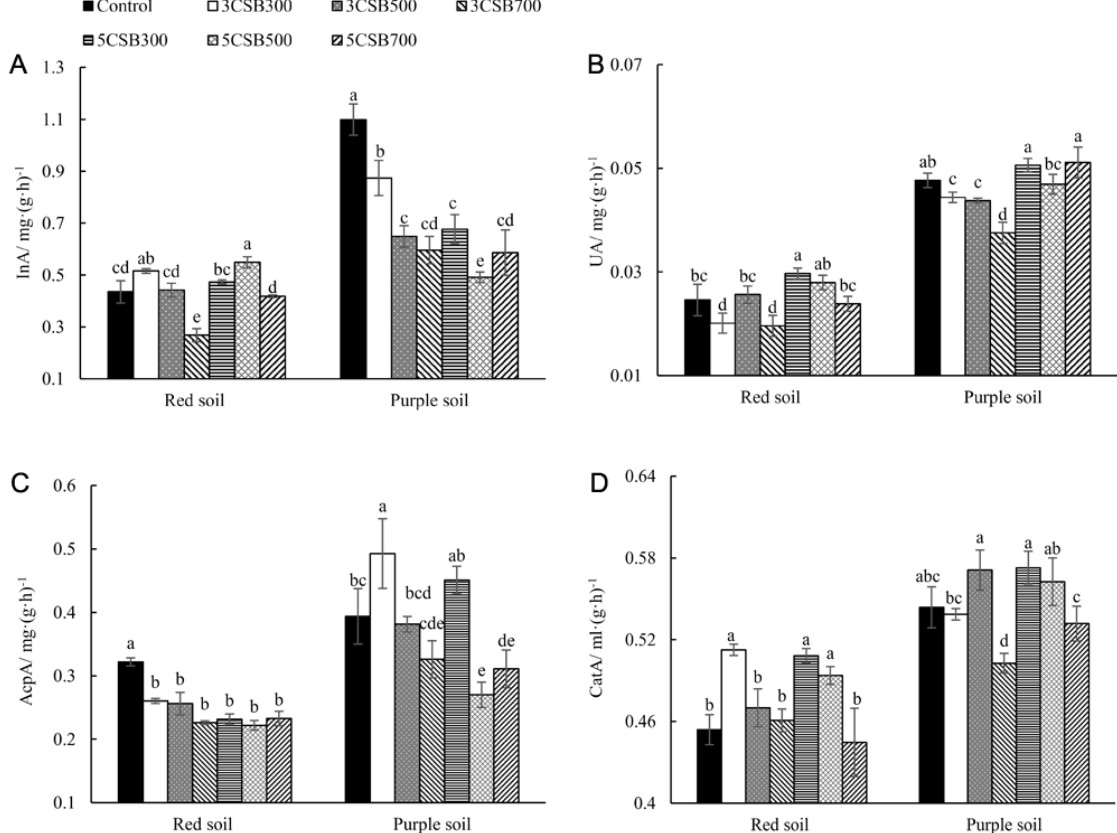

**Figure 2.** Soil enzyme activities in both soils after 180 days of incubation. The *X*-axis indicates different soil types, including red soil and purple soil. (**A**): InA activity in both soils after 180 days of incubation. (**B**): UA activity in both soils after 180 days of incubation. (**C**): AcPA activity in both soils after 180 days of incubation. (**D**): CatA activity in both soils after 180 days of incubation. Control: red/purple soil without biochar application; Abbreviation: InA: invertase activity, UA: urease activity, AcPA: acid phosphatase, CatA: catalase activity. Different lowercase letters within a line indicated significant differences between treatments at the *p* < 0.05 level. Error bars indicating S.E, (*n* = 3).

**Table 4.** Three-way ANOVA for microbial properties in both soils. The table shows *F* ratios and *P* value for the three factors: soil type (ST), biochar pyrolysis temperature (BPT) and biochar application rate (BAR).

| | | Parameters | | | | | | |
|---|---|---|---|---|---|---|---|---|
| **Effect** | | **ST** | **BPT** | **BAR** | **ST * BPT** | **ST * BAR** | **BPT * BAR** | **ST * BPT * BAR** |
| MBC | *F* | 274.7 | 83.46 | 21.86 | 14.75 | 5.69 | 8.52 | 38.87 |
| | *P*-value | <0.001 | <0.001 | <0.001 | <0.001 | 0.025 | 0.002 | <0.001 |
| MBN | *F* | 204.05 | 20.56 | 0.84 | 0.36 | 0.29 | 33.75 | 7.28 |
| | *P*-value | <0.001 | <0.001 | 0.368 | 0.705 | 0.597 | <0.001 | 0.003 |
| MBP | *F* | 445.9 | 18.38 | 15.66 | 38.47 | 22.12 | 2.16 | 11.03 |
| | *P*-value | <0.001 | <0.001 | 0.001 | <0.001 | < 0.001 | 0.137 | <0.001 |
| InA | *F* | 182.21 | 42.86 | 2.78 | 18.44 | 42.18 | 13.60 | 1.46 |
| | *P*-value | <0.001 | <0.001 | 0.108 | <0.001 | < 0.001 | <0.001 | 0.253 |
| UA | *F* | 762.30 | 8.78 | 54.36 | 1.30 | 9.81 | 3.90 | 6.14 |
| | *P*-value | <0.001 | 0.001 | <0.001 | 0.290 | 0.005 | 0.034 | 0.007 |
| AcpA | *F* | 202.48 | 32.96 | 15.87 | 23.92 | 3.92 | 4.41 | 1.14 |
| | *P*-value | <0.001 | <0.001 | 0.001 | <0.001 | 0.059 | 0.023 | 0.337 |
| CatA | *F* | 133.44 | 33.19 | 5.72 | 2.73 | 1.03 | 0.36 | 8.39 |
| | *P*-value | <0.001 | <0.001 | 0.025 | 0.086 | 0.320 | 0.700 | 0.002 |

Abbreviation: MBC: microbial biomass C; MBN: Microbial biomass N; MBP: microbial biomass P; InA: invertase activity; UA: urease activity; AcpA: acid phosphatase activity; CatA: catalase activity.

In addition, 3CSB300 and 3CSB700 treatments on the soil significantly decreased soil urease activity by 22.39% and 25.52%, respectively, compared with control in the red soil, which showed a significant inhibition of urease activity. Except for 5CSB300, there was no significant difference in urease activity in the red soil that was observed in other CSB treatments. Furthermore, 5CSB300 significantly increased urease activity in the red soil by 21.25%, compared to the control (Figure 2b). Urease activity decreased with the increase in the pyrolysis temperature of biochar application a higher application rate (5% *W/W*), and the 5CSB700 had an adverse effect on soil urease activity, but there was no difference with control and 3CSB500 treatments.

In the purple soil, CSB treatment under 3% addition rate all decreased soil urease activity significantly, dropped by 7.2–26.93%, compared to control. Urease activity was inhibited significantly with the increase in the pyrolysis temperature of biochar under lower application rate, a repressed effect also found in the treatment of 5CSB500, while 5CSB300 and 5CSB700 both promoted urease activity. Some results of previous studies reported that biochar application significantly increased urease activity, which is consistent with the treatment of 5CSB300, 3CSB500, and 5CSB500 in the red soil in our study [75–77]. Du et al. [78] and Wang et al. [79] both reported higher biochar application rates led to an increase of urease activity. However, some studies found that biochar application significantly decreased urease activity, which is consistent with our study in the red soil [80]. Bailey et al. [81] showed that biochar application to soil could increase the urease activity related to N utilization of biochar. The different results indicated that the effects of biochar on urease activity are complex and further research is required.

The significant negative effect of CSB application on soil acid phosphatase activities was observed in all CSB treatments in the red soil, i.e., decreased by 23.85–45.05%, compared to control. Moreover, there is no difference among all CSB treatments (Figure 2C). Similar results were also reported in previous studies [82–84]. After 180 days incubation, 3CSB300 and 5CSB300 resulted in an increase of the enzyme activities by 25.13% and 14.47% over control in the purple soil, respectively. However, other treatments had a negative effect on acid phosphatase activity in the purple soil, decreased by 3.30–45.93%, compared to control. Moreover, changes in soil pH value and other nutrients can also affect acid phosphatase activity in the red soil, which could confirm by the correlation analysis in our study (Table 5) [83,85]. These repressed effects may be mainly due to the excess of inorganic P provided from the biochar [55,86].

**Table 5.** Pearson's correlation coefficients of soil chemical properties and microbial properties.

|  |  | InA | Ure | AcpA | CatA | MBC | MBN | MBP |
|---|---|---|---|---|---|---|---|---|
| Red soil | pH | −0.151 | 0.321 | −0.723 ** | −0.074 | 0.215 | 0.452 * | 0.504 * |
|  | OM | 0.227 | 0.654 ** | −0.485 * | 0.517 ** | −0.496 * | −0.092 | 0.608 ** |
|  | TC | 0.125 | 0.482 * | −0.708 ** | 0.141 | 0.052 | 0.386 | 0.460 * |
|  | TN | 0.262 | 0.153 | −0.519 * | 0.177 | 0.010 | 0.300 | 0.232 |
|  | TP | −0.324 | 0.190 | −0.705 ** | −0.136 | 0.105 | 0.421 | 0.674 ** |
| Purple soil | pH | −0.839 ** | 0.129 | −0.343 | 0.020 | 0.076 | 0.289 | −0.424 |
|  | OM | −0.181 | 0.149 | 0.479 * | 0.157 | −0.532 * | 0.056 | −0.771 ** |
|  | TC | −0.804 ** | 0.015 | −0.331 | 0.204 | 0.003 | 0.317 | −0.335 |
|  | TN | 0.200 | −0.274 | 0.409 | 0.047 | −0.608 ** | 0.257 | −0.121 |
|  | TP | −0.799 ** | 0.261 | −0.512 * | 0.085 | 0.239 | 0.300 | −0.267 |

* $p < 0.05$, significant level; ** $p < 0.01$, significant level.

3CSB300, 5CSB300 and 5CSB500 significantly increased catalase activity of the red soil by 12.93%, 11.99%, and 8.77%, respectively, compared to control, although 5CSB700 decreased the catalase activity of the red soil, no difference with control and other CSB treatments (Figure 2D). In the purple soil, 3CSB700 showed a significant decrease in catalase activity, decreased by 0.04 units, while other treatments (expect for 3CSB300 and 5CSB700) showed increases of catalase activity by 0.02–0.03 units, compared to control, and no difference with control. Catalase is one of the major enzymes for indicating

the oxidation-reduction potential of soil. Masto et al. [87] and Yang et al. [88] reported that catalase activity increased significantly with an increase in the application rate of biochar. Chai et al. [77] showed that catalase activity decreased may due to fertilizer applied into the soil decreased catechol and vanillin degradation, lignocellulose depolymerization, and inhibited soil microbial metabolism capacity.

The soil microbial biomass determined by the chloroform fumigation extraction method showed a variable difference in all biochar treatments on the red and purple soil (Figure 3). In the non-CSB treatment control, the average contents of MBC, MBN, and MBP were 235.81/187.47 mg·kg$^{-1}$, 36.05/45.82 mg·kg$^{-1}$, and 0.24/1.26 mg·kg$^{-1}$ compost in the red and purple soils, respectively. On average, CSB treatments significantly affected all microbial biomass indices in soil incubation. In the non-CSB treatment control, the average contents of MBC, MBN, and MBP were 235.81/187.47 mg·kg$^{-1}$, 36.05/45.82 mg·kg$^{-1}$, and 0.24/1.26 mg·kg$^{-1}$ compost in the red and purple soils, respectively. On average, CSB treatments significantly affected all microbial biomass indices in soil incubation.

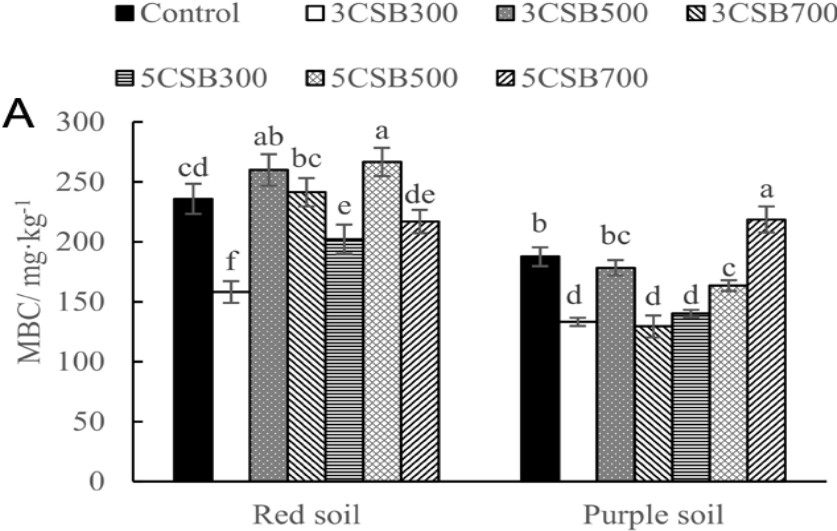

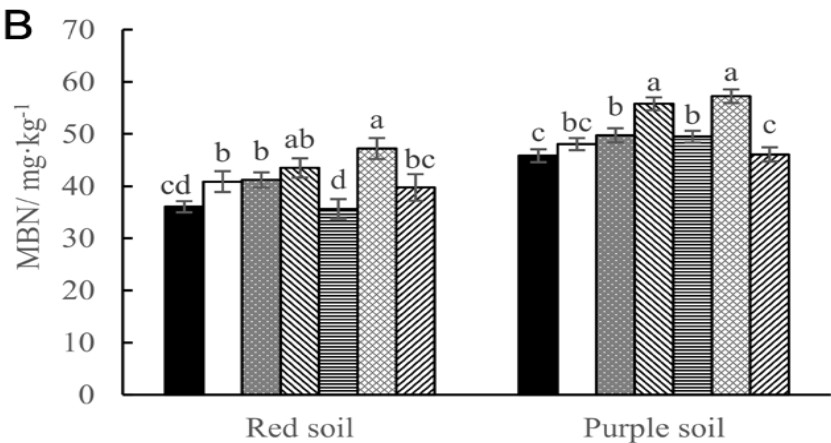

**Figure 3.** *Cont.*

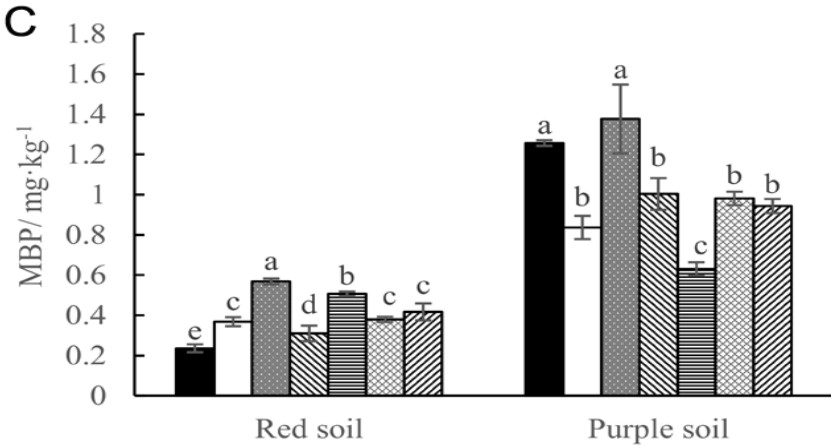

**Figure 3.** Soil microbial biomass in both soils after 180 days of incubation. The *X*-axis indicates different soil types, including red soil and purple soil. (**A**): MBC in both soils after 180 days of incubation. (**B**): MBN in both soils after 180 days of incubation. (**C**): MBP in both soils after 180 days of incubation. Control: red/purple soil without biochar application; Abbreviation: MBC represents microbial biomass carbon; MBN indicates microbial biomass nitrogen; MBP indicates microbial biomass phosphorus; Error bars indicating S.E, (*n* = 3). Different lowercase letters within a line indicated significant differences among treatments at the *p* < 0.05 level.

In the red soil, 3CSB300 and 5CSB300 led to significantly inhibited in MBC, compared to control, dropped by 77.76 and 33.67 units, respectively (Figure 3A). However, the addition of CSB500 showed a significant increase in MBC by 24.02–30.74 units. In contrast to CSB300 and CSB500, the addition of CSB700 had no significant difference with control. As for purple soil MBC, the addition of 5CSB700 led to a significant increase, compared to control and other CSB addition, increased by 31.13 units. However, other CSB treatments also showed a negative effect on purple soil MBC, which decreased by 47.34–58.15 units. The increased MBC after CSB application is consistent with previous studies, and the biochar application rate was a major factor to increase significantly the MBC, this result confirmed in the treatment of 3CSB500 and 5CSB500 in our study [89–91]. Some studies reported that the increase in MBC might due to more nutrients supplied and OM mineralized on the surface of biochar for the microbes [91,92]. Brtnicky et al. [74] reported that MBC decreased with biochar application caused by sorptive characteristics that may due to the pyrolysis temperature used in biochar produced.

The significant increases in MBN were generally between 14.26% and 30.96% stronger for CSB500 than for CSB700 between 10.24% and 20.64%, compared to control in the red soil (Figure 3B). Moreover, the addition of 3CSB300 also significantly increased MBN content by 13.27%, while 5CSB300 addition decreased by 10.34%, and no significant difference with control. Except for the addition of 5CSB700, all CSB treatments increased content of MBN in the purple soil, increased by 4.87–24.95%, compared to control, and the greatest increase in MBN in the purple soil occurred under the treatment of 5CSB500, increased by 11.43 units. The incubation experiment showed that CSB application could significantly improve the MBN content in the soil that consistent with previous studies [89,90].

In the red soil, all CSB addition showed significant positive effect on soil MBP, increasing from 0.24 mg·kg$^{-1}$ with the control treatment to 0.57 mg·kg$^{-1}$ in 3CSB500, all CSB treatments increased by 0.07–0.33 units, but there was no significant difference among the addition of 3CSB300, 5CSB500 and 5CSB700 (Figure 3C). Overall, except for 3CSB500, other CSB treatments decreased the MBP in the purple soil by 25.32–99.21%. Generally, the highest soil MBP content was observed in the purple soil under the addition of 3CSB500, which increased by 94.67%, but no significant difference with control treatment. On average, the present study suggested that the application of CSB can enhance MBN in both studied soils and MBP in red soils while had a negative effect on MBP (except for 3CSB500) in the purple soil, these CSB treated purple soils presented the significantly lower microbial activity of MBP than control (Figure 3B,C) [89,91]. Generally, there was a lower microbial activity of MBP in the red

soil than in the purple soil, this phenomenon may be due to the stronger P fixation capacity of red soil than purple soil, which means that there was less available P for microorganisms [93].

### 3.6. Relationship between Soil Chemical and Microbial Properties after 180-Day Incubation

PCA was conducted on the basis of the soil chemical properties and microbial biomass and related enzyme activity in all treatments after 180-day incubation with biochar and also showed that the microbial properties were significantly different between different treatments (Figure 4). The first component (PC1) explained 45.88% of the total variance, the PS and CSB treatments in the purple soil with positive loadings, while the RS and CSB treatments in the red soil with negative loading. The second component (PC2) explained 30.07% of the total variance, the RS, RS + 3CSB300, and RS + 3CSB500 had positive scores on PC2, other CSB treatments in the red soil had negative scores on PC2, while for the purple soil, PS and PS + 3CSB500 also had positive scores on PC2, other CSB treatment had negative scores on PC2. Furthermore, the treatments with CSB and RS in red soil were clearly distinguished from the treatments in the purple soil along PC1. These results showed that the CSB application resulted in different changes in soil chemical, enzyme activity, and microbial properties with the application rate having the most significant influence, followed by the pyrolysis temperature.

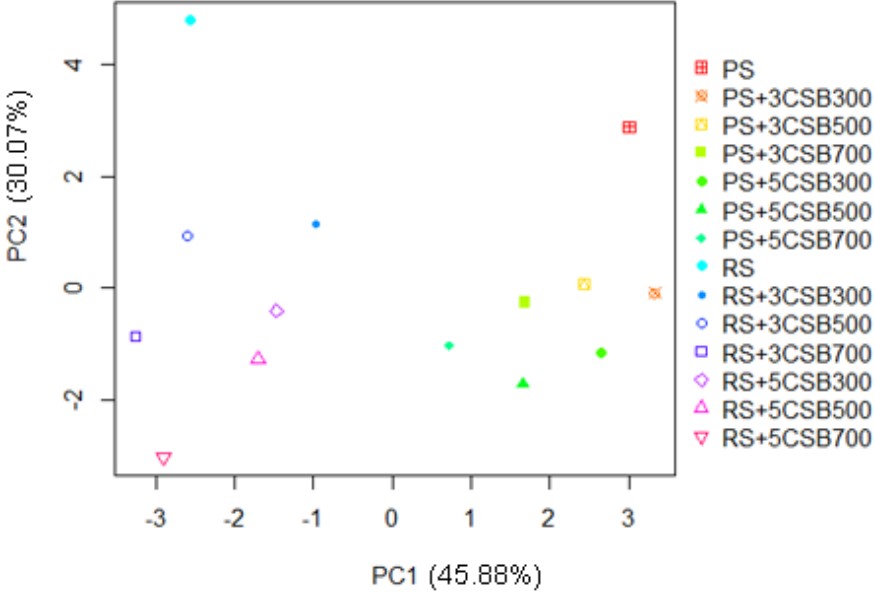

**Figure 4.** Score plot of the principal component analysis (PCA).

Pearson's correlation analysis showed that microbial properties had variable relationships with soil chemical properties in the red and purple soil (Table 4). In the red soil, pH was positively correlated with soil MBN and MBP and significantly negatively correlated with soil acid phosphatase activity, which could confirm by the other studies [83,85]. These repressed effects may be mainly due to the excess of inorganic P provided from the biochar [55,86]. Furthermore, soil OM has negatively correlated with MBC ($r = -0.496$, $p < 0.05$) and acid phosphatase activity ($r = -0.485$, $p < 0.05$), whereas positively ($p < 0.01$) correlated with urease and catalase activities. MBP was mainly dependent on the TP ($r = 0.674$, $p < 0.01$), followed by OM ($r = 0.608$, $p < 0.01$) and pH ($r = 0.504$, $p < 0.05$), TC also was positively correlated with soil MBP. However, TC and TP showed a significant negative ($p < 0.01$) correlation with acid phosphatase activity. The MBP of studied red soil generally increased in response to biochar application under all CSB treatments after 180-day incubation, these increases were mainly caused by high TP of biochar contributed to soil [55].

As for purple soil, OM and TN both showed a negative relationship with MBC ($r = -0.532$, $p < 0.05$; $r = -0.608$, $p < 0.01$), and OM also was negatively correlated with MBP ($r = -0.771$, $p < 0.01$). Soil pH, TP, and TC all showed significantly negative ($p < 0.01$) relationships with invertase activity.

The OM showed a positive relationship with acid phosphatase activity ($r = 0.479$, $p < 0.05$), while TP showed a negative ($r = -0.512$, $p < 0.05$) correlated with acid phosphatase activity.

As for enzyme activity, soil type, biochar pyrolysis temperature had a significant effect on soil invertase activity, respectively, while biochar application rate did not affect invertase activity, only showed significant effects when biochar application rate interacted with soil type or with biochar pyrolysis temperature. Soil type, biochar pyrolysis temperature, and biochar application rate all showed significant ($p < 0.05$) effects on soil urease, acid phosphatase, and catalase activity, respectively, but there were different effects when the three factors interact with each other. Especially in catalase activity, a significant effect only observed in the interaction of soil type, biochar pyrolysis temperature, and application rate (Table 5). The soil type, biochar pyrolysis temperature, and application rate on microbial biomass all showed significant effects on MBC and MBP, while biochar application rate had no significant effect on MBN, and only showed some significant effects when the interaction with biochar pyrolysis temperature or with soil type and biochar pyrolysis temperature was taken into account (Table 5).

## 4. Conclusions

In comparison with low pyrolysis temperature (300 °C) and medium pyrolysis temperature (500 °C), high pyrolysis temperature (700 °C) significantly increased soil chemical properties in both soils, especially at the higher application rate (5%). As for microbial properties, 5CSB300 significantly increased soil invertase, urease, and catalase activity in the red soil, and significantly increased soil urease, acid phosphatase, and catalase activity in the purple soil. Under biochar application at 500 °C pyrolysis temperature, soil microbial biomass (MBC, MBN, and MBP) increased significantly in the red soil, and MBN increased significantly in the purple soil. The findings of the study revealed that CSB biochar is a potential soil amendment material in acidic soils, and provided a reference for biochar-based fertilizer produced by *C. oleifera* shell for a long-term field experiment in the future. *C. oleifera* shell as a forestry residue returned to the soil in the form of biochar may cycle nutrients back into the *C. oleifera* field and even better for understanding nutrient levels in soils which is important for guiding biochar amendment use. The limitation of this study is focused on the 180-day incubation of soil–biochar mixture under a controlled condition without a plant, which differs from the field conditions, and therefore the possibilities of nutrients cycling and the interaction between nutrients cycling and plants were not taken into account. Moreover, the impacts of biochar on the ecosphere need further exploration as each participant takes its specific functional place in every special kind of environment [94]. Furthermore, this comparative experiment was based on a short-term incubation study with different pyrolysis temperatures and the application rate of biochar in the laboratory. Long-term field studies with CSB applications are needed to evaluate the longevity of the effect of CSB to *C. oleifera* field in the future study.

**Author Contributions:** Q.S., T.Z., S.C., and H.C. conceived and designed the experiments; Q.S., Y.H., and Y.W. performed the experiments; Q.S. and Y.H. analyzed the data; Q.S. wrote the manuscript. All authors read and provided comments and approved the final manuscript.

**Funding:** We acknowledge the funding from the Special Fund for Science and Technology Innovation of Fujian Agriculture and Forestry University (CXZX2017500), the Special Scientific Research Fund for Doctoral Disciplines in Colleges and Universities (20123515110010), the Fujian Science and Technology Major Project (2013NZ0001-1), the *Camellia oleifera* Germplasm Innovation and Industrialization Project (ZYCX-LY 2017003), and the Fujian Province Science and Technology Department Platform Construction Project (2010N2001).

**Conflicts of Interest:** The authors declare no conflict of interest.

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
