# Peer review of "Biochar Impacts on Acidic Soil from Camellia Oleifera Plantation: A Short-Term Soil Incubation Study"

_agronomy, doi:10.3390/agronomy10091446_

Round 1

Reviewer 1 Report

Reviewer’s comments:

This study investigated the effect of C. oleifera shell biochar (CSB) on C. oleifera plantation soils as a soil amendment. Their results showed that the soil pH, total P and available P (AP) significantly increased under the 5% application rate and 700 °C pyrolysis temperature of C. oleifera shell biochar (5CSB700) in two soils, and indicated CSB application rate and pyrolysis temperature had a significant impact on soil pH, TP, and AP (P<0.05). The manuscript is good but it needs some revision before reconsideration.

The results of your research studies are good, but authors can explain in a right way and improve clarity. All results are under a single heading? Please merge the discussion part with the results nicely to explain your findings adequately. For example, authors can explain results under following headings:

  • Characteristics of soils and CSB
  • Effect of CSB on soils chemical properties
  • Effect of CSB on soils fertility and P fractionations
  • Effect of CSB on soils microbial properties

Please use the treatment names (5CSB300) in the whole article, instead of writing lengthy statements frequently “biochar at 300C at 5% application rate.”

Add data from the recent studies. You haven’t cited studies from 2020.

Journal name is missing in reference no. 42 to 92. Add the Journal names, volume and page no. where missing.

Entire Manuscript: Many of the sentences in the article contain spelling mistakes, or some are unclear. Therefore, the entire article needs to be revised thoroughly.

Discussion section is weak and must be improved based on relevant new literature.

Introduction is not logical and revise it to establish novelty of study.

Author Response

Dear reviewer,

Thank you for your comments.

Please see the attachment for the response.

Kindly regards,

Reviewer 2 Report

- The writing of the manuscript should be improved. Consider consulting with a editor or rewriting the manuscript.

- In table 1, what do you mean by the yield of biochar?

- Line 128, the word ph should be written like pH.

- What was the design? CRD, RDB?

- Line 126, pH has a logarithmic value so the increase in pH should be reported logarithmic.

- The names of 5CSB300 and 3CSB300 are confusing to me. I recommend mentioning 3% CSB300 biochar rather than 3CSB300.

- Line 175, remove the word data from the sentence.

- Line 182, the last sentence doesn’t have a verb. I recommend adding, “were not reported” to the end of the sentence.

- Line 184, the font (P <0.05) 
is different.

- Consider changing the numbers of the tables to one digit and make the tables look less crowded.

- Mention what is the X-axis under the figure 2 and 3.

- Line 247-262: consider shortening this part.

- Results should be compared with other recent related studies.

- Figures should be discussed in the discussion too. Figure 4 was not discussed.

- At the end there should be a suggestion to test the biochar in the field and the recommend it for long term uses.

Author Response

Dear reviewer,

Thank you for your comments.

Please see the attachment for the responses.

Kindly regards,

Round 2

Reviewer 1 Report

I am happy to accept revised manuscript.

Author Response

Thank you for your comments.